# Isolation and Bioassay of a New Terminalone A from *Terminalia arjuna*

**DOI:** 10.3390/molecules28031015

**Published:** 2023-01-19

**Authors:** Khalil ur Rehman, Dilfaraz Khan, Abdulrahman A. Almehizia, Ahmed M. Naglah, Asma S. Al-Wasidi, Moamen S. Refat, Mohamed Y. El-Sayed, Hamid Ullah, Shafiullah Khan

**Affiliations:** 1Institute of Chemical Sciences, Gomal University, Dera Ismail Khan 29050, KPK, Pakistan; 2Drug Exploration and Development Chair (DEDC), Department of Pharmaceutical Chemistry, College of Pharmacy, King Saud University, Riyadh 11451, Saudi Arabia; 3Department of Chemistry, College of Science, Princess Nourah bint Abdulrahman University, Riyadh 11671, Saudi Arabia; 4Department of Chemistry, College of Science, Taif University, P.O. Box 11099, Taif 21944, Saudi Arabia; 5Chemistry Department, College of Science, Jouf University, P.O. Box: 2014, Sakaka, Saudi Arabia; 6Department of Chemistry, Balochistan University of Information Technology, Engineering and Management Sciences, Quetta 87300, Pakistan

**Keywords:** *Terminalia arjuna*, Terminalone A, flavonoids, antibacterial, antioxidants, reactive oxygen specie

## Abstract

*Terminalia arjuna* possesses significant cardioprotective, antidiabetic and antioxidant properties as these properties are described in Ayurveda. In the present study, three flavonoids were isolated through the separation and chromatographic purification of the whole plant material of *T. arjuna*. Spectroscopic characterization identified one of them as a new flavonoid “*Terminalone* A (**1**)” and two known flavonoids i.e., *6-hydroxy-2-(4-hydroxyphenyl)-7-methoxy-4H-chromen-4-one* (**2**) and *2-(3,4-dihydroxyphenyl)-5,7-dihydroxy-4H-chromen-4-one* (**3**). The bioactivity studies showed considerable antibacterial and antioxidant (DPPH radical scavenging) potential for all the three compounds **1**–**3** where the compound **1** showed strong antibacterial and antioxidant activity.

## 1. Introduction

The *Terminaliaarjuna* (*Roxb. ex DC.*) Wight & Arn. plant belongs to the second biggest genus “*Terminalia*” of the *Combretaceae* family possessing more than 200 species. It is commonly recognized as *Arjuna* [1]. Terminalia species are widely distributed in humid and hot places of Asia, Australia and Africa [2]. The height of the arjuna tree is about 60 to 80 feet. Among the various medicinally important Asian-based species, *T. arjuna* and *T. chebula* are enormously reported in the literature as well as recognized for great medicinal uses in Ayurvedic [3]. The plant usually develops in a wide range of soil; however, the most suitable is sticky soil and red lateritic soil [4]. It is an evergreen tree, and it develops new leaves usually from February to April [5]. The plant possesses the shape of a spreading crown and has dropping branches; therefore, it is planted for ornamental and shade purposes [6].

The *T. arjuna* plant is locally named as “*Arjun*”, famous as the garden’s protector, and therefore also recognized as “baaghon ka muhafiz” in the local community of Pakistan [7]. *T. Arjuna* is reported for its diverse range of bioactivities including antibacterial, antioxidant, anti-inflammatory and antitumor. It also possesses hypolipidemic, antimutagenic and hypocholesterolemic effects [8]. Ayurvedic doctors recommend *T. arjuna* in the treatment of three kinds of tumors, i.e., Kapha, Vata and Pitta [2]. It has been reported that Saponin glycosides obtained from *T. Arjuna* are accountable for the inotropic impacts. The plant has also been used for cancer treatment and as cardioprotective, where the flavonoids isolated from the plants are suggested to be responsible for these cardiotonic and anticancer properties [9]. Powder from the bark stem of *T. arjuna* has been used against coronary infection and hypercholesterolemia [10].

The *T. arjuna* plant is well-recognized for having different classes of medicinally important natural products, e.g., it contains high contents of polyphenolic compounds (60–70%) including ellagic acid, arjunolone, arjunone, gallic acid, oligomeric proanthocyanidins and others [5]. It also contains triterpenoids, saponins, Tannins, Phytosterols and minerals such as Ca, Zn, Mg and copper. Furthermore, the *T. arjuna* plant is rich in Amino acids such as Tryptophan, Cysteine, Tyrosine and Histidine [9]. Among the various natural products, flavonoids reported from this plant have been recognized for diverse biological activities including antioxidant and antibacterial [11]. Therefore, the present study describes the isolation of a new phenolic compound (**1**), along with two known phenolic metabolites (**2**, **3**) from *T. arjuna*, assesses their antibacterial and antioxidant potential, and studies the mechanistic features of the bacterial inhibition. This study is based on chromatographic purification and isolation of phenolic compounds from the whole plant materials and a correlation between these phenolic metabolites and antioxidant and antibacterial potential.

## 2. Results and Discussion

In the present work, compounds **1**–**3** were initially isolated from EtOAc soluble fraction of methanolic extract of *T. Arjuna*, then structural elucidation of the obtained compounds was carried out using modern spectroscopic techniques and finally, the compounds were evaluated for their antioxidant and antibacterial potential.

### 2.1. Characterization of the Isolated Compounds

Repeated column chromatographic techniques were applied to the selected ethyl acetate soluble fraction of the *Terminalia Arjuna* plant material which provided compounds **1**–**3** as brown solids. Their structures were established by using modern spectroscopic techniques such as mass spectrometry, IR, ^1^H and ^13^C NMR. The isolated compounds **2** and **3** were known compounds, and by comparison of their spectroscopic data with the reported literature, they were identified as *6-hydroxy-2-(4-hydroxyphenyl)-7-methoxy-4H-chromen-4-one* (**2**) [11] and *2-(3,4-dihydroxyphenyl)-5,7-dihydroxy-4H-chromen-4-one* (**3**) (see Figure 1) [12]. The new compound *Terminalone* A (**1**) was obtained as a sticky brown solid. The FTIR absorption peaks observed at 3320–3310 cm^−1^ and 1605 cm^−1^, 1343 cm^−1^ and 1155 cm^−1^ suggested that hydroxyl moiety and benzene ring are present in compound **1 [13]**. Its molecular formula was established through HREI-MS as C_28_H_36_O_16_ that displayed the [M + H]^+^ peak at *m/z* 629.2072 (calcd. for C_28_H_36_O_17_ as 629.2076), indicating eleven degrees of unsaturation, which represents the presence of five rings, including two benzene rings. Additionally, the linked scanning on a molecular ion (M^+^) indicated that the ions with m/z 465, 287 and 195 arose directly from it, corresponding to the fragments M-C_6_H_11_O_5_ (M-163) with the loss of one glucose unit, M-C_12_H_21_O_11_ (M-241) with loss of two glucose units and M-C_18_H_25_O_12_ (M-433) with the loss of two glucose units along with the ring B, suggesting the presence of two glucose units [14]. The ^1^HNMR spectra of compound **1** showed a singlet at *δ* 8.35 ppm (s) confirming the presence of the methin proton at position 6 of ring A. A pair of doublets (d) at *δ* 7.25 ppm (*J* = 7.3 Hz) and *δ* 6.61 ppm (*J* = 1.7 Hz) and a doublet of the doublet signal at *δ* 6.59 (*J* = 7.3, 1.7) of ring B, confirming a 1′, 3′, 4′ tri-substituted B-ring (Table 1, Note: See the table at the end of this document). A couple of doublets at *δ* 3.03/2.63 ppm are attributed to the methylene proton of ring C. The four-hydroxyl moiety of rings (a), (b) and (c) resonated at *δ* 3.35–3.64 ppm as a singlet. The eight oxymethines resonated at around 3.57 to 3.22 ppm, two oxomethylenes signals at *δ* 4.56 dd (*J* = 1.8, 12.2, H_b_)/3.78 dd (5.2, 12.2, H_a_) and 4.31 (dd, *J* = 2.2, 12.0 Hz, H_b_)/3.61 (dd, *J* = 5.0, 12.0 Hz, H_a_), and the two anomeric peaks at *δ* 4.96 (d, *J* = 7.2) and 4.90 (d, *J* = 7.6 Hz, H-1″) suggested the presence of two pyranose moieties [15]. The ^13^C NMR spectra with the help of DEPT (distortion-less enhancement via polarization transfer) indicated signals for 1 CH_3_, 3–CH_2_-, 16–CH- and eight quaternary carbons (Table 1, Note: See the table at the end of this document). The highly downfield peaks at *δ* 132.5, 130.0 and 127.7 ppm were assigned to the phenolic Ar carbon, while *δ* 115.03 ppm and *δ* 132.5 ppm were attributed to Ar-quaternary carbon (C-4a to C-8a). Other signals appearing at *δ* 97.3 ppm and *δ* 95.6 ppm correspond to anomeric Carbon-1″ and Carbon-1′′′, while the peaks for other carbons of glucose moiety resonated in the chemical shift region from 83.2–64.3 ppm, respectively. The obtained ^1^HNMR and ^13^CNMR data for compound **1** are given in Table 1 and Appendix A and have close agreement with the reported literature [14,16]. Based on these findings, compound 1 was assigned the name Terminalone A.

### 2.2. Antibacterial Property of Compounds **1**–**3**

The three isolated natural products **1**–**3** were tested against *E. coli* and *S. aureus.* These bacteria are very toxic and spread infection in different parts of humans and animals. The results of all the three compounds are shown in Table 2.

The results illustrated that compound **1** has much higher inhibition against both bacteria as compared to compounds **2** and **3**. The potency of compound **1** against *S. aureus* is much higher than that of *E. coli*, which may be due to the strong and complex cell membrane of *E. coli*. The zones of inhibition (mm) of compounds **1**–**3** against *E. coli* and *S. aureus* were 18.4 (±0.3), 13.5 (±0.4) and 11,7 (±0.4) and 21.7 (±0.4), 15.6 (±0.3) and 14.3 (±0.5), respectively. Streptomycin was also applied against both bacteria as a standard drug. The result showed that the standard drug has a much higher inhibition efficiency against both the bacteria with the inhibition zone of 23.5 (±0.3), 26.4 (±0.4) against *E. coli* and *S. aureus* respectively. Furthermore, the disruption to *E. coli* in the presence of compound **1** was also examined through scanning electron microscopy as shown in Figure 2C. It is clear from the result that the morphology of *E. coli* is completely damaged. Most of the bacteria have shrunk, and the cell membrane was entirely ruptured.

#### 2.2.1. Minimum Inhibitory Concentrations (MIC) of Compounds **1**–**3** and Streptomycin

MIC is the lowest concentration of a compound that inhibits the growth of bacteria. MIC of compounds **1**–**3**, and streptomycin was examined in this assay. Different concentrations for all the compounds (15–120 µg/mL) were examined against *E. coli* and *S. aureus*. In all these compounds, streptomycin showed the lowest MIC value against both bacteria. In all the compounds (**1**–**3**), compound **1** showed better MIC value against *E. coli* and *S. aureus*. The MIC values for all the three compounds against *E. coli* were found to be 60, 75 and 105 µg/mL and against *S. aureus* 45, 75 and 75 µg/mL, respectively. The MICs of streptomycin standard and compounds **1**–**3** are presented in Table 3.

#### 2.2.2. Mechanism of Inhibition of the Selected Pathogens and Examination of Reactive O-Species (ROS)

Considering mechanistic aspects of bacterial inhibition, the formation of reactive oxygen species can be correlated to the antimicrobial impact of compound **1** in the microbial cell. The reactive O-species such as •O_2_¯, •OH and H_2_O_2_ are released because of the energetic electrons of the OH bunch in compound **1**, which lengthens the age of ROS within the *E. coli* cell. Such reactive species are considered to be very harmful to the DNA and protein of *E. coli*. Furthermore, it has been demonstrated that the 2,7-dichlorofluorescein diacetate color oxidized into its dichlorofluorescein derivatives when exposed to ROS. Additionally, green fluorescence was examined upon excitation at about 488 nm in the presence of compound **1** as displayed in Figure 2B. In the absence of compound **1**, no fluorescence was noticed as shown in Figure 2A. These findings strongly endorse that the reactive O-species are delivered in the bacteria cells in the presence of compound **1**, which is accountable for their inhibition. Subsequently, these findings suggest that compound **1** has an association with the *E. coli* cells’ surface which is consistent with the formation of intracellular reactive O-species and spillage of cytoplasm as illustrated in Figure 2C.

### 2.3. DPPH Scavenging Activity of Compounds **1**–**3**

DPPH scavenging assays of all the three compounds were examined and compared with vitamin C (Vit. C) as shown in Figure 3. DPPH was selected for this study because it is a very stable radical due to its resonance structure. The results demonstrated that the radical scavenging efficacy of all the three compounds improved with enhancing their concentrations. The scavenging activity of compound 1 is more proficient than those of compounds 2 and 3, which may be due to a lot of phenolic OH present in compound 1. The phenolic OH has the ability to donate electrons and stabilize the radicals. The results showed that more than 90% scavenging was achieved at a dose of 0.8 mg/mL of compound 1. On the other hand, compounds 2 and 3 enabled DPPH scavenging of 77% and 66%, respectively, at 0.8 mg/mL concentrations.

## 3. Experimental

### 3.1. General Procedure

Silica-gel-coated, aluminum-support-based TLC cards (60 F_254_, 0.2 mm thick; E. Merck, Darmstadt, Germany) were applied to check the purity of the compounds. Silica gel (230 to 400 mesh, E. Merck) was used in chromatographic columns for purification. Glass-supported preparative thin layer chromatographic plates (20 × 20, 2 mm) coated with silica gel (0.5 mm thickness) were used to purify the semi-pure compounds. Solutions of Cerium (IV) sulfate [Ce(SO_4_)_2_] and Potassium permanganate [KMnO_4_] were used for visualization of the compounds on TLC plates. Ethanol was used in recording UV spectrum through UV spectrophotometer (Shimadzu, Japan, UV-2700) spectrophotometer. A Shimadzu-460 IR spectrometer was employed to record the IR spectra of the isolated constituents. The TOF Mass spectrometer (Billerica, MA, USA) was used for mass spectral studies of the isolated compounds. The ^1^H and ^13^CNMR spectrums of the obtained compounds were recorded by using a Bruker 300 MHz (300 MHz for ^1^H and 75 MHz for ^13^C) NMR spectrometer where DMSO-d_6_ was used as solvent.

### 3.2. Plant Materials

The plant in whole was collected from the different areas of Gomal University. Then, Professor Saddiq Khan from the Faculty of Agriculture (Gomal University) identified the plant, where a voucher specimen had been deposited in an herbarium (Accession No. C-0046).

### 3.3. Extraction and Isolation of the Compounds

The *T. arjuna* whole plant material (15 kg) was subjected to methanolic (3 × 50 L of MeOH) extraction at room temperature. The extract gummy residue of about 600 g was obtained upon evaporation under reduced pressure. The obtained residue was then suspended in H_2_O and further extracted with different solvents such as n-hexane, dichloromethane, EtOAc and MeOH to get 92 g, 112 g, 76 g and 65 g of the corresponding fractions, respectively. The ethyl acetate soluble fraction was then loaded over silica gel packed columns using different eluting systems in order of increasing polarity starting from *n*-hexane (non-polar), following through *n*-hexane-DCM (with an increase of 10% polarity stepwise), DCM-ethyl acetate (with an increase of 10% polarity stepwise) and EtOAc-MeOH (with an increase of 5% polarity stepwise) system. Various fractions were obtained, and the fractions that displayed similar results on TLC were combined, which resulted in seven sub-fractions. The sub-fraction AF displayed two spots on the TLC card with significant Rf value; therefore, a preparative TLC method was adopted to purify the compounds in each spot using *n*-hexane: EtOAc (8:2–3:7) as eluting solvents. The bands of the corresponding spots were scratched from the TLC plates and were filtered where fine brown solids were obtained that were latterly recognized as compound **2** (12 mg) and **3** (14 mg). The sub-fraction AS, which produced four major spots on TLC along with some minor spots, was further loaded over silica gel CC eluted with n-Hex: EtOAc (with an increase of 5% polarity stepwise), which resulted in the further four sub-fractions. The sub-fraction ASL, on keeping under a fume hood, rapidly changed into fine brown solid with *n*-hexane: EtOAc (4:6). This solid material was then identified as compound **1** (9 mg) with the help of spectroscopic techniques, Appendix A. The pure constituents were screened for antibacterial and antioxidant activity using standard protocols.

### 3.4. Antibacterial Activity of Compounds **1**–**3**

The agar well diffusion procedure was followed to determine the antibacterial efficacy of compounds **1**–**3**. Two potential drug-resistant *E. coli* (ATCC 6538) and *S. aureus* (ATCC 6633) bacteria were selected for this study. The said bacteria were cultured in broth media and incubated at 37 °C for 15 hrs. The inocula of each bacterium were splashed on the agar media. Sterile cork borer made two wells of 6 mm. About 60 μL of 0.8 mg/mL of each compound was poured in the wells. Afterward, the discs were incubated for 24 h at 37 °C. Streptomycin was used as a standard.

#### 3.4.1. MIC of Compounds **1**–**3** and Streptomycin

MIC of all the compounds was investigated by the already reported serial dilution method [17]. Various sterile test tubes having 1 mL of the said bacterial solution and different concentrations or dilutions (15–120 μg/mL) of compounds were taken and incubated for 24 h in a shaking incubator at 37 °C. The test tube having only bacterial solution was taken as control.

#### 3.4.2. Test for Reactive O-Species (ROS) in the Presence of Compound **1**

2,7-dichlorodihydrofluorescein diacetate color was used to check the production of reactive O-species. This important hue provides a very accurate assessment of ROS in bacterial cells. Specific concentrations of compound **1** and *E.coli* were incubated for three hours at 300 rpm. After appropriate incubation, the *E. coli* strain suspension was precipitated for eight min at 7000 rpm, then the obtained pellet was rinsed with a saline buffer of phosphate. The phosphate saline buffer including the pellet was accordingly blended in with 1 mL of 15 mM of 2,7-dichlorodihydrofluorescein diacetate color for sixty min. The color-treated cells were subsequently washed with phosphate saline buffer in order to eliminate the dye from the surface of the cells. The fluorescence magnifying instrument was utilized to capture fluorescence pictures at excitation and emission frequencies i.e., at 488 nm and at 535 nm separately [18].

### 3.5. Antioxidant Activity

The antioxidant potential of the isolated compounds **1**–**3** was examined according to the reported method [19]. Initially, 2 mM of DPPH in combination with various concentrations of the isolated compounds (0.1–0.8 mg/mL) was constantly stirred. These were then kept in the dark for about 30 min. After that, the absorbance was studied by UV–Vis spectrophotometer at 517 nm. Vitamin C at the same concentration as that of test compound (0.1–0.8 mg/mL) was used as a control. The % inhibition was examined by the following formula, Equation (1).
(1)$$\%\;\mathrm{inhibition}\;=\frac{\left(\mathrm{Absorbance}\;\mathrm{by}\;\mathrm{the}\;\mathrm{control}\right)-\left(\mathrm{Absorbance}\;\mathrm{by}\;\mathrm{the}\;\mathrm{sample}\;\mathrm{compounds}\right)\;}{\left(\mathrm{Absorbance}\;\mathrm{by}\;\mathrm{the}\;\mathrm{control}\right)}$$

## 4. Conclusions

This study demonstrated that Terminalone A **1**, a new flavonoid, along with two known flavonoids (6-hydroxy-2-(4-hydroxy phenyl)-7-methoxy-4H-chromen-4-one **2** and 2-(3,4-dihydroxy phenyl)-5,7-dihydroxy-4H-chromen-4-one **3,** were attained from the EtOAc soluble fraction of *T. arjuna*. Further bioassay investigation of the isolated products recognized them as significant antibacterial and antioxidant agents where the new compound **1** showed highest activity such as 18 (±0.3) against *E. coli* and 21 (±0.4) against *S. aureus.* Overall, the present research shows agreement with the reported ethnomedicinal importance of *T. arjuna* and exploited its hidden medicinal properties.

## Figures and Tables

**Figure 1 molecules-28-01015-f001:**
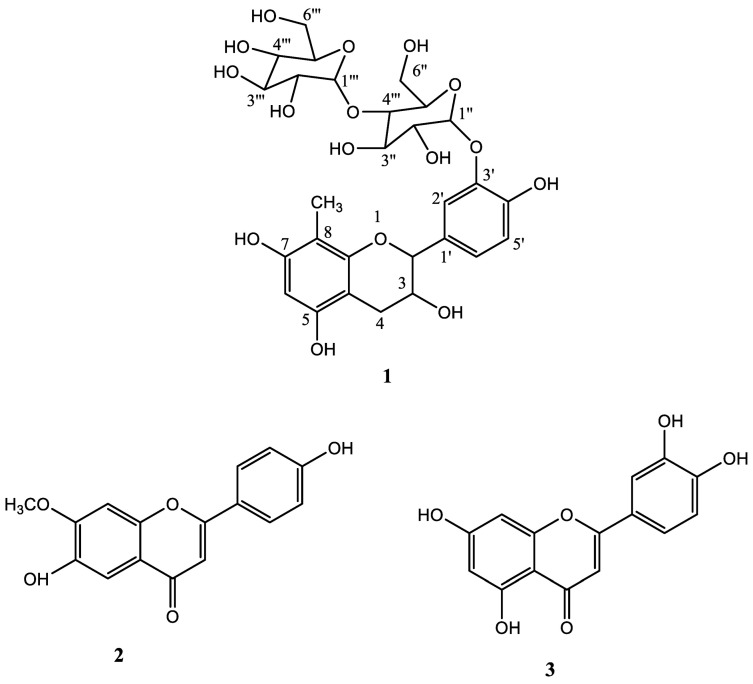
Structures of Isolated Compounds (**1**–**3**).

**Figure 2 molecules-28-01015-f002:**
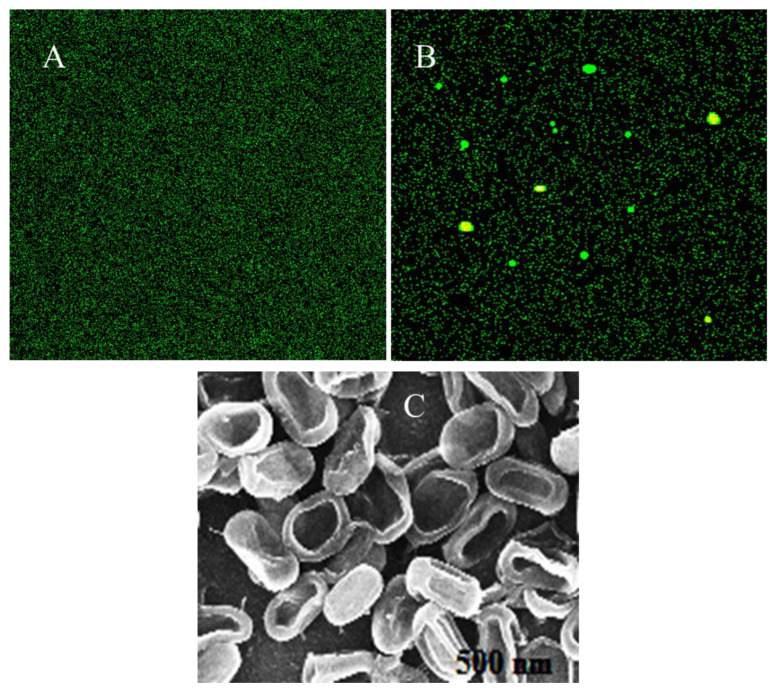
ROS examination (**A**) in the absence of compound **1**, (**B**) in the presence of compound **1** and (**C**) scanning electron microscopic examination of *E. coli* in the presence of compound **1**.

**Figure 3 molecules-28-01015-f003:**
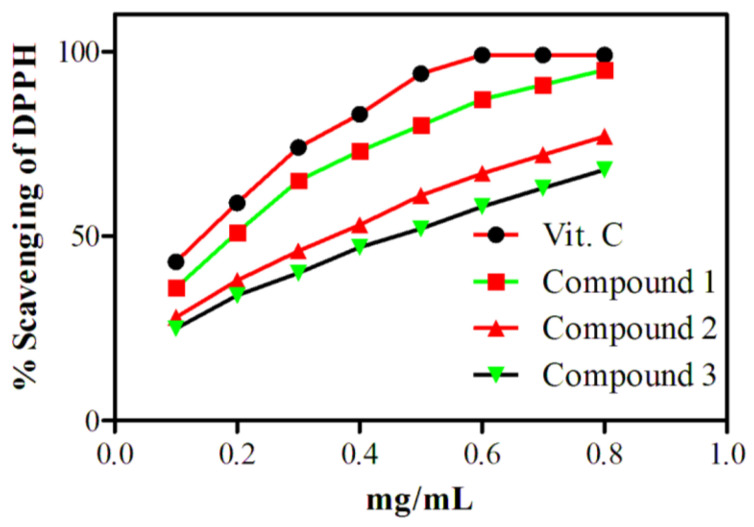
DPPH scavenging activities of compounds **1–3**.

**Table 1 molecules-28-01015-t001:** ^1^H NMR and ^13^C NMR of compound **1** (CDCl_3_, δ ppm).

H/C	^1^H NMR (*δ* ppm)	^13^C NMR (*δ* ppm)
2 (CH)	5.38 (d, *J* = 6.7 Hz)	69.7
3 (CH)	4.89 (m)	63.5
4 (CH_2_)	3.01 (dd, *J* = 6.4, 15.2 Hz)/2.63 (dd, *J* = 2.1, 15.2 Hz)	39.56
4a	Q	115.03
5	Q	127.7
6 (CH)	8.35 s	98.4
7	Q	130.0
8	Q	104.6
8-Me	2.51 s	29.4
8a	Q	132.5
1′	Q	102.4
2′ (CH)	6.61 (d, *J* = 1.7))	111.3
3′	Q	129.2
4′	Q	120.01
5′ (CH)	7.25 (d, *J* = 7.3)	103.4
6′ (CH)	6.59 (dd, *J* = 7.3, 1.7 Hz)	100.0
1″ (CH)	4.96 (d, *J* = 7.2 Hz)	97.3
2″ (CH)	3.24 (dd, *J* = 7.2, 9.0 Hz)	76.1
3″ (CH)	3.35 (dd, *J* = 9.0, 7.3 Hz)	83.2
4″ (CH)	3.40 (dd, *J* = 8.6, 7.3 Hz)	73.4
5″ (CH)	3.57 (m)	79.2
6″ (CH_2_)	4.56 (dd, *J* = 1.8, 12.2, H_b_)/3.78 (dd, *J* = 5.2, 12.2, H_a_)	65.6
1″′ (CH)	4.90 (d, *J* = 7.6 Hz)	95.6
2″′ (CH)	3.22 (dd, *J* = 7.6, 8.6 Hz)	75.0
3″′ (CH)	3.27 (dd, *J =* 8.6, 7.2 Hz)	81.9
4″′ (CH)	3.38 (t, *J* = 8.0, 7.2 Hz)	70.1
5″′ (CH)	3.50 (m)	77.2
6″′ (CH_2_)	4.31 (dd, *J* = 2.2, 12.0 Hz, Hb)/3.61 (dd, *J* = 5.0, 12.0 Hz, Ha)	64.3

**Table 2 molecules-28-01015-t002:** Antibacterial activity of the isolated natural products **1**–**3** and streptomycin.

Bacteria	Antibacterial Activity (Inhibition Zone in mm)			
	Compound 1	Compound 2	Compound 3	Streptomycin
*E. coli*	18.4 (±0.3)	13.5 (±0.4)	11.7 (±0.4)	23.5 (±0.3)
*S. aureus*	21.7 (±0.4)	15.6 (±0.3)	14.3 (±0.5)	26.4 (±0.4)

**Table 3 molecules-28-01015-t003:** MIC of standard drug and compound **1**–**3** against *E. coli* and *S. aureus*.

Compounds	MIC (µg/mL)		
	*E. coli*	*S. aureus*	
Standard (streptomycin)	30		15<
Compound **1**	60		45
Compound **2**	75		75
Compound **3**	105		75

## Data Availability

Not applicable.

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
