# Peer review of "Isolation and Bioassay of a New Terminalone A from Terminalia arjuna"

_molecules, 2023, doi:10.3390/molecules28031015_

Round 1

Reviewer 1 Report

In the reports by Bushra et al, have reported Isolation and Bioassay of a New Terminalone A from Terminalia arjuna. Author uses modern techniques such as NMR and FTIR to evaluate the chemical structure of isolated compound. Overall, this is a clear, concise, and well-written manuscript. I have some suggestion before publishing.

Comments:

1.     Author should identify the plant Terminalia arjuna with accession number of specimen from herbarium or any reputed institution which provide identification number.

2.     Authors should include more references in discussion section of NMR and FTIR related to this work which support the study.

3.     Authors have to added the peaks illustration graph of NMR and FTIR.

4. Authors have added more significance of this study in the introduction part.

Author Response

Reviewer 1 Comments and Suggestions for Authors

In the reports by Bushra et al, have reported Isolation and Bioassay of a New Terminalone A from Terminalia arjuna. Author uses modern techniques such as NMR and FTIR to evaluate the chemical structure of isolated compound. Overall, this is a clear, concise, and well-written manuscript. I have some suggestion before publishing.

  1. Author should identify the plant Terminalia arjunawith accession number of specimen from herbarium or any reputed institution which provide identification number.

Response to Reviewer 1: Dear reviewer, we are grateful to you to take the time to review our MS and pointed such golden comments. The accession number and herbarium name was provided in the manuscript as suggested.

  1. Authors should include more references in discussion section of NMR and FTIR related to this work which support the study.

Response to Reviewer 1: The authors thank reviewer for their fruitful comments and suggestions. References were added in the discussion section of the manuscript accordingly.

  1. Authors have to added the peaks illustration graph of NMR and FTIR.

Response to Reviewer 1: Added in the supplementary file as per suggestion.

  1. Authors have added more significance of this study in the introduction part.

Response to Reviewer 1: The authors value the reviewer suggestions and at the end of introduction some more descriptions were added to make it significant.

Reviewer 2 Report

The manuscript addresses an interesting aspect with promising results. However, it wasn’t matching the standards of a scientific paper. There are many points the authors should be considered to improve their manuscript:

·         The writing style was poor and required improvement, as well as many grammatical mistakes, which should be avoided.

·         The experimental section should be included a thorough explanation of each step of the experiment using simple scientific language.

·         The authors didn’t mention why they excluded compounds 2 and 3 from the test of reactive O-species (ROS).

·         The discussion part was very weak and lack of referring to previous studies that related to the topic. 

Author Response

Reviewer 2 Comments and Suggestions for Authors

The manuscript addresses an interesting aspect with promising results. However, it wasn’t matching the standards of a scientific paper. There are many points the authors should be considered to improve their manuscript:

  1. The writing style was poor and required improvement, as well as many grammatical mistakes, which should be avoided.

Response to Reviewer 2: Thanks to reviewer for giving a time to our manuscript from the busy schedule and provided us an opportunity to polish our manuscript by giving valuable comments and suggestions. The manuscript was thoroughly checked and any grammatical and typographical mistakes found anywhere were corrected accordingly.

  1. The experimental section should be included a thorough explanation of each step of the experiment using simple scientific language.

Response to Reviewer 2: The authors thank reviewer for such a valuable comments and suggestions. Special attentions were given to the experimental section and was polished and refined as per suggestion.

  1. The authors didn’t mention why they excluded compounds 2 and 3 from the test of reactive O-species (ROS).

Response to Reviewer 2: Compound 1 being a new compound was our only target of interest. The other two compounds (2 and 3) were known compounds and their data is already reported therefore they were excluded from the tests.

  1. The discussion part was very weak and lack of referring to previous studies that related to the topic.

Response to Reviewer 2: We have taken a reviewer's suggestion and references were added in the discussion portion in order to correlate each data with the previous literature. 

Reviewer 3 Report

1. Line 65 . Solutions of Ce(SO4)2 and  KMnO4 were used for visualization of the compounds on TLC plates.

Kindly give the full name for reagent used

2. The other two compounds already reported.

3. The antibacterial activity already reported.

The work is very less the author just reported isolation of one compound. The yield is not reported for compound 1.

Additional experiment needed

Author Response

Reviewer 3 Comments and Suggestions for Authors

  1. Line 65. Solutions of Ce(SO4)2and KMnO4 were used for visualization of the compounds on TLC plates.Kindly give the full name for reagent used.

Response to Reviewer 3: Thanks to reviewer for giving a time to our manuscript from the busy schedule and provided us an opportunity to polish our manuscript by giving valuable comments and suggestions. Full names for these chemicals were added accordingly.

  1. The other two compounds already reported.

Response to Reviewer 3: Thanks, the compounds 2 and 3 are already reported and reported literature for these compounds have been cited with referred in the manuscript.

  1. The antibacterial activity already reported.

Response to Reviewer 3: We concentrated over the mechanistic study of the antibacterial activity exhibited by these compounds. Furthermore, our main concentration was toward the activity of the new compound (which is not reported) and its comparison with the reported compound with the reported compounds and with the standard drug.

  1. The work is very less the author just reported isolation of one compound. The yield is not reported for compound 1.

Response to Reviewer 3: The authors thank reviewer for their fruitful comments. The yields of the new compound along with the known compounds were mentioned in the manuscript.

Reviewer 4 Report

The manuscript describes the isolation, characterization of one compound and studies of the biological activity of three compounds. The language is very poor. I believe that it is impossible to isolate single compounds from plant material using classical column chromatography. NMR spectra for the isolated compounds are not shown. If you assign signals to individual atoms also it is necessary to register two-dimensional spectra and show them. Two glucose residues are shown in the formula of compound 1. In contrast, the text mentions rhamnose. Rhamnose has a methyl group instead of a hydroxymethyl group. The description of the IR spectrum is very poor, the spectrum is not shown. The discussion describing the determination of the structure of the compound 1 by NMR is unconvincing, especially when it comes to the structure of the disaccharide. The labeling of atom numbers in the figure showing the structure is missing. There is no comment at all on how the structure of compounds 2 and 3 was determined. They are not glycosides like compound 1. This seems to be impossble The description of biological studies contains a lot of hypotheses that have not been verified. The results reported along with uncertainties are incorrectly presented. The result and uncertainty must have the same decimal expansion. This is not the case in this manuscript. For new compounds, in addition to high-resolution mass spectrometry, elemental analysis should be done. It is missing. 

Author Response

Reviewer 4 Comments and Suggestions for Authors

The manuscript describes the isolation, characterization of one compound and studies of the biological activity of three compounds.

  1. The language is very poor.

Response to Reviewer 4: Dear reviewer, we are grateful to you for taking the time to review our MS and pointed such golden comments. The language was polished to the best of our knowledge and the grammatical and typographical mistakes found anywhere were corrected.

  1. I believe that it is impossible to isolate single compounds from plant material using classical column chromatography.

Response to Reviewer 4: Dear reviewer, after loading several times to get the sub-fractions each time and to apply series of chromatographic techniques again and again it is possible to isolate the pure compounds through column chromatography. However, their yield is very low because most of the weight losses occur during each lading, because most of the polar metabolites stick with the polar silica gel of the column.

  1. NMR spectra for the isolated compounds are not shown.

Response to Reviewer 4: Thanks, The NMR spectrum for the new compound has been included in the supplementary materials. The data of known compounds were compared with the literature and cited in the manuscript.

  1. If you assign signals to individual atoms also it is necessary to register two-dimensional spectra and show them.

Response to Reviewer 4: The authors value reviewer suggestions. The compounds were obtained with very low yield. Only 9 mg of the new compound was obtained, which was utilized for 1D NMR spectra as well as for the activity and IR data. Therefore, the remaining was not enough for to get the 2D NMR spectra,

  1. Two glucose residues are shown in the formula of compound 1. In contrast, the text mentions rhamnose. Rhamnose has a methyl group instead of a hydroxymethyl group.

Response to Reviewer 4: Thanks, the glucose residues are pyranose rings, and hence the mentioned mistake was corrected in the manuscript accordingly.

  1. The description of the IR spectrum is very poor; the spectrum is not shown. The discussion describing the determination of the structure of the compound 1 by NMR is unconvincing, especially when it comes to the structure of the disaccharide.

Response to Reviewer 4: The authors thanks reviewer for their valuable comments. The IR data was described and explained clearly in the manuscript and also the data was referred and correlated with the previous literature and cited.

  1. The labeling of atom numbers in the figure showing the structure is missing.

Response to Reviewer 4: Thanks, the atoms were labeled with numbers accordingly.

  1. There is no comment at all on how the structure of compounds 2 and 3 was determined. They are not glycosides like compound 1. This seems to be impossible.

Response to Reviewer 4: The identification of the structure of known compounds were explained in the manuscript, and they were identified by comparison of their spectroscopic data with the reported literature. The reported literature has been cited in the manuscript.

  1. The description of biological studies contains a lot of hypotheses that have not been verified. The results reported along with uncertainties are incorrectly presented. The result and uncertainty must have the same decimal expansion. This is not the case in this manuscript.

Response to Reviewer 4: The mentioned corrections were done and the results and the uncertainty were brought to the same decimal expansions as suggested.

  1. For new compounds, in addition to high-resolution mass spectrometry, elemental analysis should be done. It is missing.

Response to Reviewer 4: Thanks, the HREI-MS data that gave the molecular ion peak has been mentioned in the manuscript that displayed the [M+H]+ peak at m/z 629.2072 (calcd. for C28H36O17 as 629.2076).

Round 2

Reviewer 2 Report

The manuscript has been improved; however, it wasn't sufficient to be accepted. 

Reviewer 3 Report

Kindly correct grammer.

Reviewer 4 Report

No comments.